# Synergistic Autophagy Effect of miR-212-3p in Zoledronic Acid-Treated In Vitro and Orthotopic In Vivo Models and in Patient-Derived Osteosarcoma Cells

**DOI:** 10.3390/cancers11111812

**Published:** 2019-11-18

**Authors:** Ju Yeon Oh, Eun Ho Kim, Yeon-Joo Lee, Sei Sai, Sun Ha Lim, Jang Woo Park, Hye Kyung Chung, Joon Kim, Guillaume Vares, Akihisa Takahashi, Youn Kyoung Jeong, Mi-Sook Kim, Chang-Bae Kong

**Affiliations:** 1Laboratory of Biochemistry, School of Life Sciences and Biotechnology, Korea University, Seongbuk-gu, Seoul 136-701, Korea; ojo5295@kirams.re.kr (J.Y.O.); joonkim@korea.ac.kr (J.K.); 2Division of Radiological Science and Clinical Translational Research Korea Cancer Center Hospital, Nowon-gu, Seoul 01812, Korea; 3Department of Biochemistry, School of Medicine, Daegu Catholic University, Nam-gu, Daegu 42472, Korea; eunhokim8@gmail.com (E.H.K.); sunha112@cu.ac.kr (S.H.L.); 4Division of Radiation Biomedical Research, Korea Institute of Radiological and Medical Sciences, Seoul 01812, Korea; eyeonjoo@gmail.com; 5Department of Basic Medical Sciences for Radiation Damages, National Institute of Radiological Sciences, Chiba 263-8555, Japan; sai.sei@qst.go.jp; 6Korea Drug Development Platform using Radio-isotope, Korea Institute of Radiological & Medical Sciences, Seoul 139-706, Korea; jangwoo@kirams.re.kr (J.W.P.); hkchung@kirams.re.kr (H.K.C.); 7Cell Signal Unit, Okinawa Institute of Science and Technology Graduate University (OIST), Okinawa 1919-1, Japan; guillaume.vares@oist.jp; 8Gunma University Heavy Ion Medical Center, Maebashi 371-8511, Gunma, Japan; a-takahashi@gunma-u.ac.jp; 9Research Center for Radiotherapy, Korea Institute of Radiological and Medical Sciences, Seoul 139-706, Korea; amy3523@kirams.re.kr; 10Department of Radiation Oncology, Korea Institute of Radiological and Medical Sciences, Seoul 139-706, Korea; 11Department of Orthopaedic Surgery, Korea Institute of Radiological and Medical Sciences, Seoul 139-706, Korea

**Keywords:** zoledronic acid, osteosarcoma, autophagy, miR-212

## Abstract

Osteosarcoma (OS) originates from osteoid bone tissues and is prone to metastasis, resulting in a high mortality rate. Although several treatments are available for OS, an effective cure does not exist for most patients with advanced OS. Zoledronic acid (ZOL) is a third-generation bisphosphonate that inhibits osteoclast-mediated bone resorption and has shown efficacy in treating bone metastases in patients with various types of solid tumors. Here, we sought to clarify the mechanisms through which ZOL inhibits OS cell proliferation. ZOL treatment inhibited OS cell proliferation, viability, and colony formation. Autophagy inhibition by RNA interference against Beclin-1 or ATG5 inhibited ZOL-induced OS cell death. ZOL induced autophagy by repressing the protein kinase B/mammalian target of rapamycin/p70S6 kinase pathway and extracellular signal-regulated kinase signaling-dependent autophagy in OS cell lines and patient-derived OS cells. Microarrays of miRNA showed that ZOL increased the levels of miR-212-3p, which is known to play an important role in autophagy, in OS in vitro and in vivo systems. Collectively, our data provided mechanistic insight into how increased miR-212-3p through ZOL treatment induces autophagy synergistically in OS cells, providing a preclinical rationale for conducting a broad-scale clinical evaluation of ZOL + miR-212-3p in treating OS.

## 1. Introduction

Osteosarcoma (OS) arises from osteoid tissues in the bone, and most often localizes in the metaphysis of adolescent long bones. OS is characterized by a high propensity for metastasis, resulting in a high incidence of death [1]. Current treatments for OS include surgery, radiation therapy, chemotherapy, and other recently developed treatments, such as immunotherapy and targeted therapy. Unfortunately, there is a lack of effective cures for most patients with advanced OS.

Currently, bisphosphonates represent the most important class of inhibitors of osteoclast-mediated bone resorption [2,3] and are used extensively for treating skeletal diseases such as Paget’s disease [3,4], postmenopausal osteoporosis [3,5], and tumor-induced osteolysis [3,6]. Bisphosphonates are pyrophosphate analogs that bind to hydroxyapatite, accumulate in bones, and inhibit osteoclastic activity [3]. Zoledronic acid (ZOL), a third-generation nitrogen-containing bisphosphonate, is an inhibitor of osteoclast-mediated bone resorption with confirmed efficiency in treating bone metastases in cancer patients with breast [3,4,5,6], prostate, lung, and other solid tumors [6].

Recent data from in vitro studies also indicate that ZOL decreases OS cell proliferation by activating the immune system, suppressing angiogenesis, and inducing apoptosis [7]. Interestingly, the use of ZOL can also overturn drug resistance in OS [8]. Moreover, ZOL has the ability to lower primary tumor growth, inhibit lung metastasis, and extend survival in animal models of OS [9]. In a four-patient-cohort study, following initial treatment of high-grade OS with ZOL, the median progression-free survival extended to 19 months, and the median overall survival increased to more than 56 months [10]. Previous studies have shown that ZOL exerts antitumor activity in several human neoplasms, including myeloma and breast, prostate, colon, and pancreatic cancers via mechanisms associated with growth factor release, cell adhesion, and apoptosis [11]. The clinical benefits of ZOL have been extended to patients with bone metastases secondary to a broad range of solid tumors including prostate cancer, lung cancer, and renal cell carcinoma [12]. ZOL is the current clinical standard for preventing bone metastasis of human cancers [13].

Autophagy, the process of degradation and regeneration of cellular components, not only protects cells, but also induces apoptosis [14]. One of the main functions of autophagy in cancer is tumor suppression, which is regulated by the autophagy-related (*Atg*) gene [15]. In mammals, the autophagy-inducible Beclin-1 complex containing class III phosphatidylinositol 3-phosphate kinase controls autophagosome initiation [16] and plays an important role in autophagy. Therefore, autophagy is a valid target for cancer treatment because it inhibits tumors. In OS, autophagy is one of the mechanisms involved in chemotherapy-induced cell death [17].

miR-212-3p, a tumor-associated miRNA, has been reported to suppress many types of cancer. miR-212-3p serves as a tumor suppressor in non-small cell lung cancer (NSCLC) [18] and gastric carcinoma [19]. However, data from other studies show that miR-212-3p has oncogenic properties in colorectal cancer [20], prostate cancer [21], and pancreatic cancer [22]. Therefore, the biological functions of miR-212-3p are cancer-type specific and partly reflect the different cellular contexts of various tumors. However, to the best of our knowledge, the role and molecular mechanism of ZOL-induced miR-212-3p in OS has not been determined. In this study, we aimed to investigate the expression levels and underlying regulatory mechanism of action of miR-212-3p induced by ZOL in OS tissues and cell lines and its clinical significance in regulating OS cell death and autophagy.

## 2. Results

### 2.1. ZOL Decreased OS Cell Proliferation in a Dose-Dependent Manner

Cell viability decreased in a dose-dependent manner after ZOL exposure (Figure 1a). ZOL significantly suppressed KHOS/NP of Human Caucasian osteosarcoma Isolated from transformed foci of human osteogenic sarcoma (HOS) cells and U2OS cells of originally known as the 2T line, proliferation, as evidenced by reduced clonogenic survival (Figure 1b). As shown in Figure 1c, a significantly lower percentage of 5-bromo-2′-deoxyuridine (BrdU)-positive cells was observed in ZOL-treated cells than in control cells. Flow cytometry and TUNEL assays results showed that ZOL-treatment increased apoptotic cell death (Figure 1d,f). Consistent with this finding, we observed an increased expression of Cleaved-caspase3 in ZOL-treated cells (Figure 1g). To confirm changes in the proliferation status of tumors, cells from the orthotopic tumor model were stained for Ki67. Cells from control orthotopic tumor model mice showed stronger Ki67 positivity than cells from ZOL-treated mice (Figure 1e). These results showed that apoptosis increased after ZOL treatment. 

### 2.2. ZOL Induced Accumulation of Acidic Vacuoles (AVOs)

Figure 2a shows representative examples of both ZOL-treated and ZOL-untreated cell lines. After a 48-h treatment with 40 μM ZOL, the number of visible vacuoles in malignant cells increased significantly. In contrast to the control cells, the ultrastructures of Giemsa-stained KHOS/NP and U2OS cells treated with ZOL (for up to 48 h) showed morphological changes throughout the cytoplasm and in the cell membrane, including the loss of plasma membrane integrity and obvious vacuole formation. This marked vacuolization of the cytoplasm (without an apparent loss of nuclear material) was consistent with the known macrostructure of cells undergoing autophagy. Because ZOL induced vacuole formation, we next performed fluorescence-activated cell sorting (FACS) analysis of acridine orange (AO)-stained AVOs using ZOL-treated cells. Based on the study by Kadowaki and Karim [24], we used the red-to-green fluorescence ratio as an indicator of AVO accumulation, and therefore of autophagic progression. Quantification of AVOs revealed an increase in AVOs in ZOL-treated KHOS/NP cells, U2OS cells, and OS patient cells (Figure 2b). Treatment with 40 μM ZOL (up to 48 h) led to 3.65- and 5.75-fold increases of AVOs in KHOS/NP and U2OS cells, respectively, compared to that in control cells; the bright red fluorescence intensity also increased (*y*-axis) in the cell types. As shown in Figure 2c, ZOL-treated cells exhibited an accumulation of large autophagic vacuoles with a typical double-layer membrane and organelle remnants, whereas only a few vacuoles were observed in control cells. The Cyto-ID dye accumulated in ZOL-treated KHOS/NP and U2OS cells, and in patient-derived OS cells as like autophagy inducer Rapamycin (Figure 2d,e).

### 2.3. ZOL Treatment Induced Autophagy

Autophagy is associated with the modification of LC3B-I to a membrane-bound form, LC3B-II, which is relocated to autophagosomal membranes during autophagy [25]. Representative fluorescence micrographs showed a punctate pattern of LC3B-II expression (Figure 3a). After 48-h treatment with ZOL (40 μM), a marked elevation in the number of cells with visibly increased punctate fluorescence was observed, particularly in the peri-nuclear region of the cytoplasm.

ZOL treatment greatly increased the conversion of LC3-I to LC3-II (LC3-II:LC3-I ratio) at the protein level in KHOS and U2OS cells (Figure 3b). Moreover, we observed an increased expression of Beclin-1, which plays a vital role in regulating the early stages of autophagosome formation (Figure 3c). The Band intensities for target proteins were normalized to that for β-actin (Appendix A). Consistent with this finding, the results also showed that ZOL induced autophagy in primary OS cells derived from patients (Figure 3d,e). The Band intensities for target proteins were normalized to that for β-actin (Appendix A). Mouse xenograft cells were stained and immunoblotted with anti-LC3 antibodies to clarify whether ZOL could induce autophagy in vivo (Figure 3f). The results showed that LC3 expression increased in the ZOL-treated orthotopic model compared to that in the control group.

### 2.4. Inhibition of ZOL-Induced Autophagy Decreased Cell Death

As shown in Figure 4a, knockdown of either Beclin-1 or ATG5 expression significantly restored the proliferation of ZOL-treated cells. We next inhibited autophagy with LY294002 (20 μM) or 3-methyladenine (3-MA; 2 mM), both of which are widely used as autophagy inhibitors. Inhibiting autophagy yielded results similar results to those obtained in 3-(4,5-dimethylthiazol-2-yl)-2,5-diphenyltetrazolium bromide (MTT) and trypan blue cell-viability assays (Figure 4b,c), indicating that both compounds significantly prevented OS cell death induced by ZOL. These data are consistent with the morphological changes. Figure 4d shows that pre-treatment with 3-MA decreased the number of autophagic markers (vacuoles) and dead cells. 

### 2.5. ZOL Suppressed the mTOR/p70S6K Signaling Pathway

The Akt/mTOR/p70S6K signaling pathway is a well-studied pathway that negatively regulates autophagy [26]. Therefore, we examined the effect of ZOL on this pathway. Western blotting was performed to investigate whether mTOR and Akt activities were affected by ZOL. After a 48 h treatment with ZOL, the cells exhibited markedly lower mTOR phosphorylation at Ser2448 and Akt phosphorylation at Ser 473 (Figure 5a). Inhibition of mTOR in these cell lines is likely to affect downstream signaling molecules such as p70S6K and 4E-BP1. We therefore examined p70S6K and 4E-BP1 levels after ZOL treatment. Production of mTOR target proteins, 4E-BP1 and p70S6K, was highly reduced following ZOL treatment (Figure 5b). ZOL treatment for up to 48 h resulted in complete inhibition of 4E-BP1 phosphorylation at Thr37/46 (Figure 5b). Consistently, control xenograft cells showed stronger *p*-Akt positivity than ZOL-treated orthotopic tumor model cells (Figure 5c). Extracellular signal-regulated kinase (Erk1/2) is known to regulate the expression of autophagy and stimulate autophagy by interacting with LC3 [27]. Recent studies showed the unconventional functions of autophagy (ATG) proteins and LC3-II in upregulating Erk phosphorylation [28]. We found that decreased Erk1/2 phosphorylation (*p*-Erk1/2-T202/Y204) was observed in OS cells treated with ZOL for 48 h (Figure 5a).

### 2.6. miR-212-3p Directly Targeted and, Thus, Positively Regulated Autophagy after ZOL Treatment

As shown in Figure 6a, the miRNA array data showed that miR-212-3p was significantly upregulated, with a 2-fold increase in ZOL-treated OS cells. Among the dysregulated miRNAs, we confirmed that treatment with ZOL (40 μM) increased miR-212-3p levels in OS cells by qRT-PCR (Figure 6b). Importantly, Kaplan–Meier analysis revealed that sarcoma patients with low miR-212-3p levels displayed a higher recurrence rate than patients with high miR-212-3p expression (Figure 6c). Additionally, primary tumor cells from patients showed downregulated miR-212-3p expression, as determined by qRT-PCR (Figure 6d). Combination treatment significantly inhibited cell growth, as determined by performing trypan blue-exclusion and MTT assays in two OS cell lines and patient-derived OS cells (Figure 6e,f). As shown in Figure 6g, an increased amount of the Cyto-ID green autophagy dye accumulated in ZOL + miR-212-3p mimic-treated KHOS/NP and U2OS cells and patient-derived OS cells, compared with cells treated with ZOL or miR-212-3p mimic alone. The results showed that ZOL-induced autophagy was increased by treatment with ZOL + miR-212-3p mimic and that LC3 expression significantly increased (Figure 6h). Combined treatment increased the accumulation of Cyto-ID (Figure 6i), and these data were consistent with the morphological changes observed by Giemsa staining (Figure 6j).

### 2.7. The Autophagic Effects of Combined miR-212-3p and ZOL Treatment in In Vivo Orthotopic Model

Based on in vitro findings, to explore whether this phenotype can be exploited therapeutically, we examined the effect of miR-212-3p in nude mouse models of human OS assigned to one of five groups: Control; ZOL only; miR212-3p mimics only; ZOL + miR212-3p mimics; and ZOL + miR212-3p inhibitor. Importantly, when ZOL was combined with miR-212-3p mimics, there was a significant reduction in tumor growth as compared with miR-212-3p mimic-treated mice or with ZOL-treated mice, and miR-212-3p inhibitor treated mice recovered the inhibitory effect (Figure 7a). As shown in Figure 7b, low uptake of [Fluorine-18(18F)]-fluorodeoxyglucose (FDG) was observed in tumors treated with ZOL + miR212-3p as compared to tumors receiving either of the treatments. To confirm successful transfection, qRT–PCR analysis of miR-212-3p was performed on OS tissues (Figure 7c). Consistently, tumor weight was reduced in combination-treated mice as compared to single treated group (Figure 7d). Notably, miR-212-3p administration did not produce any significant behavioral changes or weight loss in treated animals. No pathologic features were detected by histological analysis of tissues, including the liver and spleen of treated mice, clearly indicating absence of acute toxicity (Figure 7e and Appendix A). The mRNA level of Beclin-1 increased following exposure to ZOL combined miR212-3p treatment in OS cells (Figure 7f). Next, we attempted to confirm the association between ZOL + miR212-3p and autophagy in the OS in vivo orthotopic model. As shown in Figure 7g, immunohistochemistry (IHC) analysis demonstrated that the ZOL + miR212-3p combination group significantly elevated LC3 and Beclin-1 expression levels and suppressed *p*-mTOR expression level compared to controls. 

## 3. Discussion

Chemotherapy is an important part of treatment for most OS patients, although it may not be necessary for some patients with low-grade OS. Almost all OS patients are treated with chemotherapy before surgery as neoadjuvant chemotherapy for approximately 10 weeks and then treated after surgery as adjuvant chemotherapy for up to one year. People with high-grade OSs who respond well to chemotherapy before surgery usually receive the same chemotherapeutic drugs after surgery. Thus, it is necessary to identify novel targeted drugs to improve the therapeutic outcomes in OS patients. Recently, it has been well established that autophagy induction can provide therapeutic benefits and that modulators of autophagy may serve as novel therapeutic therapies, which may ultimately lead to new therapeutic strategies against cancer.

ZOL has been found to exhibit antitumor and anti-proliferative effects [7]. In our previous study, ZOL induced apoptosis in OS cells by increasing the expression levels of cleaved PARP, caspase-3, and Bax, and combination treatment with ZOL with IR increased the efficiency of radiation therapy in OS patient-derived cells [23]. In this study, to define its mechanism of action via other cell-death pathways in addition to apoptosis, we showed that ZOL induced autophagy in OS cell lines (as in vivo orthotopic tumor models) and in OS patient-derived cells (through the Akt/mTOR/p70S6K and Erk signaling pathways), and that autophagy suppression may attenuate the anticancer effect of ZOL. However, the mechanism underlying this regulatory process needs to be elucidated in greater detail.

Recently, autophagy was observed to play a key role in tumor suppression [29], and data from a previous study indicated that inhibiting autophagy is a promising approach for cancer therapy [30]. Many signaling pathways that regulate autophagy and apoptosis are altered in cancer cells. However, the mechanism of ZOL-induced autophagy is unclear, and exactly which signaling molecules and pathways are involved warrants further exploration. In this study, LC3B-localization experiments revealed that ZOL-treated cells have typical autophagic morphology and biochemical signatures. The autophagic effects of ZOL were evident based on drug-induced Beclin-1 and Atg5 expression, together with the conversion of LC3B-I to LC3B-II in time- and concentration-dependent manners. mTOR/Akt/p70S6K signaling can halt and regulate cellular catabolic processes such as autophagy, which is frequently dysregulated in cancer and metabolic disorders [31]. Moreover, it has been reported that Akt negatively regulates autophagy by activating mTOR, which inhibits various autophagy-promoting proteins via phosphorylation [32]. We characterized the effect of ZOL on the mTOR/Akt/p70S6K pathway in this study and found that ZOL markedly inhibited mTOR phosphorylation at Ser2448 and Akt phosphorylation, which prevents the activation of mTOR and Akt signaling. Thus, p70S6K and 4E-BP1 inhibition may directly result from the upstream inhibition of mTOR and Akt by ZOL. Erk signaling is another pathway associated with autophagy that regulates various events of cell physiology, including autophagy. It has been reported that growth factors increase the interaction of Erk cascade components with autophagy proteins in both the cytosol and the nucleus [28]. Moreover, the same treatment decreased Erk phosphorylation (*p*-Erk1/2-T202/Y204), as determined by Western blot analysis. Therefore, it is necessary to further confirm the effect of the Erk phosphorylation inhibitor U0126, to demonstrate that inhibition of Erk phosphorylation downregulates LC3-II levels, thereby inhibiting autophagy. These results indicate that both the Akt/mTOR and Erk signaling pathways were involved in autophagy induction by ZOL in OS cells.

Whether autophagy enhances cell death or enhances survival remains controversial [33]. Although drug-induced autophagic tumor cell death has been reported [34], the results from most studies confirm the survival role of autophagy in chemotherapy-induced cell death [32]. Autophagy is thought to vary, depending on the cell type, cell-cycle phase, genetic background, and microenvironment [35]. One question that arises is what role autophagy plays in ZOL-treated OS cells. To directly address this question, we inhibited OS-induced autophagy pharmacologically with LY294002 and 3MA in OS cells. We found that ZOL-induced cell death significantly decreased upon the treatment with LY294002 and 3MA. Moreover, microscopy-based analysis showed that ZOL in combination with LY294002 or 3MA induced lesser but clear morphological changes in terms of cell death versus ZOL treatment alone, indicating that ZOL-induced autophagy might play a role in OS cell death.

Several chemotherapeutic agents have been shown to induce autophagy in human OS cells [36]. Certain anticancer therapeutic agents target pathways involved in autophagy, including dihydroartemisinin, which is reported to inhibit the nuclear translocation of nuclear factor-κB [37]; thiazolidinedione, which induces autophagy in breast cancer cells by activating peroxisome proliferator-activated receptor-γ [38]; curcumin, which suppresses the growth of malignant gliomas by inducing autophagy through a mechanism mediated by the Akt and Erk signaling pathways [39]; and E platinum, which induces autophagy by inhibiting mTOR phosphorylation in BGC-823 gastric carcinoma cells [40]. In addition to anticancer agents (doxorubicin, temozolomide, and etoposide), histone deacetylase inhibitors, imatinib, and anti-estrogen hormonal therapy have also been shown to induce autophagy [41]. These results suggested that basal autophagy is crucial for suppressing spontaneous tumorigenesis. Thus, we suggest that ZOL functions as a targeted drug that induces autophagy in OS cells, as evidenced by the results of one patient in this study.

miR-212 is an important miRNA and has been studied in many kinds of cancers, including hepatocellular carcinoma and pancreatic cancer [22]. In this study, we investigated miR-212 expression in ZOL-treated cells for the first time and found that its expression was markedly higher in ZOL-treated patient tissues than in the adjacent normal tissues. Importantly, the low level of miR-212 significantly increased after combination treatment with ZOL and other agents. These results suggested that miR-212 plays an important role in OS cell autophagy.

Traditionally, autophagy has been reported as a mechanism of cell survival and cell death [17]. Autophagy can contribute to both chemosensitivity and chemoresistance in OS therapy [17]. These two opposing mechanisms of autophagy appear to vary according to the experimental conditions and are not clearly defined [42]. Our findings showed that autophagy inhibits tumor growth by suppressing the mTOR/p70S6K pathway, and previous studies confirmed the tumorigenic effect of autophagy using doxorubicin and cisplatin, which inhibit tumor growth [17,43]. Previous research revealed that miR-212-3p negatively controls starvation-induced autophagy in PCa cells by suppressing SIRT1 and that mir-212-3p upregulation led to the suppression of angiogenesis and cellular senescence [44]. In this study, mirR-212-3p overexpression after ZOL treatment significantly suppressed cell proliferation, increased autophagy, and markedly promoted OS cell apoptosis. Thus, restoring the expression of tumor-suppressive miRNAs seems to be a promising strategy for cancer treatment [45]. Using mimics for miR-34a and let-7 [46] or lentiviral vectors encoding let-7 [47] achieved remarkable therapeutic benefits against both murine and human NSCLC. Thus, miR-212-3p mimics or viral vectors encoding miR-212-3p may also emerge as viable treatment options for OS in the future. Moreover, the results of our in vitro and in vivo experiments suggested that miR-212-3p may become a therapeutic target for OS. Therefore, comparing the advantage of miR-212-3p targeted therapies to conventional and targeted drugs (e.g., ZOL) for OS will be of great importance in further studies. We should confirm the molecular mechanisms in the miR-212-3p-mediated enhancement of ZOL-induced autophagy, such as the possible target genes for miR-212-3p. Also, there is increasing evidence for cell death by autophagy and apoptosis. Especially, developmental cell death in autophagic cancer cell death increased by novel anti-cancer drugs. In our results, understanding the interplay between apoptosis and autophagy in cancers needs to identify new molecular targets and other signaling pathway for cancer therapy.

In conclusion, drug-induced, miRNA-regulated autophagy is increasingly becoming a therapeutic strategy for managing treatment of cancer patients. Our findings provide insights into the anticancer efficacy of ZOL and miR-212-3p, supporting a preclinical rationale for conducting a large-scale clinical study on treating OS.

## 4. Materials and Methods

### 4.1. Cell Culture and Tissue Samples

The KHOS/NP and U2OS OS cell lines were purchased from the American Type Culture Collection (Rockville, MD, USA) and used in this study. The OS cell lines were cultured in Dulbecco’s Modified Eagle Medium (WelGene, Daegu, Korea) containing 10% (*v*/*v*) fetal bovine serum (FBS; Gibco^®^, Life Technologies, Carlsbad, CA, USA) and 1% (*v*/*v*) penicillin-streptomycin (Gibco^®^, Life Technologies). OS tissue was obtained with informed consent from a patient who underwent surgery at the Korea Institute of Radiological and Medical Sciences (Institutional Review Board Approval Number K-1603-001-001), and a primary cell culture was established from this tissue as previously described [48].

### 4.2. Reagents

Antibodies against Atg5, Beclin 1, LC3(A/B), p-mTOR (Ser2448), p-Akt (S473), p-P70S6K (Thr389), p-4EBP1 (Thr37/46), and p-p44/42 MAPK (Erk1/2) (Thr202/Tyr204) were purchased from Cell Signaling Technology (Danvers, MA, USA). Antibodies against beta-actin and Ki67 were purchased from Santa Cruz Biotechnology (Dallas, TX, USA). ZOL was purchased from Sigma–Aldrich (St. Louis, MO, USA). Two autophagy inhibitors, namely the PI3K inhibitor LY294002 and the autophagosome inhibitor 3MA, were obtained from Sigma–Aldrich.

### 4.3. MTT Assay

After ZOL treatment, cell viability was assayed via MTT (Sigma–Aldrich) testing.

### 4.4. Colony-Formation Assay

Cells were treated with ZOL for 48 h and then incubated for 7 days. The resulting colonies were fixed with 100% methanol for 30 min and stained with 0.4% crystal violet (Sigma–Aldrich) and then counted. 

### 4.5. Detection of Apoptotic Cells by Annexin V Staining

ZOL was added to the cells, which were incubated for a further 72 h. The cells were washed with phosphate-buffered saline (PBS), trypsinized, and re-suspended in 1× binding buffer (10 mM HEPES/NaOH [pH 7.4], 140 mM NaCl, and 2.5 mM CaCl_2_) at a density of 1 × 10^6^ cells/mL. Aliquots (100 μL) of the cell suspension were mixed with 5 μL annexin V FITC (PharMingen, San Diego, CA, USA) and 10 μL propidium iodide stock solution (50 μg/mL in PBS) by gentle vortexing, followed by a 15-min incubation at room temperature in the dark. Buffer (400 μL, 1×) was added to each sample and analyzed on a FACScan flow cytometer (Becton Dickinson, San Jose, CA, USA). A minimum of 10,000 cells was counted for each sample, and data analysis was performed using CellQuest software (BD Biosciences, San Jose, CA, USA).

### 4.6. Western Blotting

ZOL was added to the OS cells, which were then incubated for 48 h. The cells were then lysed with Cell Lysis buffer (Cell Signaling Technology). Proteins were separated by sodium-polyacrylamide gel electrophoresis and transferred to nitrocellulose membranes. The membranes were blocked with 1% (*v*/*v*) non-fat dried milk in Tris-buffered saline with 0.05% Tween-20 and incubated with the indicated antibodies. Primary antibodies were used at a 1:1000 dilution, and secondary antibodies were used at a 1:5000 dilution. Immunoreactive protein bands were visualized by enhanced chemiluminescence (Thermo Fisher Scientific, Waltham, MA, USA) and scanned.

### 4.7. Morphology 

To examine the effect of ZOL on cell morphology, ZOL-treated cells were stained with Giemsa. Briefly, cells seeded in six-well plates were allowed to adhere overnight onto cover slips followed by ZOL treatment. The cells were fixed with methanol for 10 min and stained with Giemsa (10% in PBS) for 15 min, followed by washing with tap water. Images were acquired using a Nikon Eclipse Ts2R-FL microscope.

### 4.8. 5-Bromo-2′-Deoxyuridine (BrdU)-Labeling Assay

BrdU-labeling assays were performed in 96-well plates using the BrdU Cell Proliferation Assay Kit (Cell Signaling Technology). After ZOL treatment, 10 μmol/L BrdU was added to each well, and the cells were incubated for 12 h at 37 °C. BrdU signaling was determined using a Multiscan FC ELISA reader (Thermo Fisher Scientific) at 450 nm.

### 4.9. Orthotopic Model and Histological Analysis

Twelve 4-week-old female BALB/c nude mice (average weight: 12.1 g, range: 11.3–13.1 g) were purchased from ORIENT Bio (Seoul, Korea) and quarantined for 1 week prior to experimentation. KHOS/NP orthotopic tumors were established as previously described [23].

### 4.10. Quantification of Acidic Vacuoles (AVOs) by Acridine Orange (AO) Staining

Cells were treated with this concentration of ZOL (40 μM) for the indicated time points, followed by staining with 1 μM AO for 15 min. Cells were then washed, re-suspended in PBS, and subjected to fluorescence-activated cell sorting (FACS) analysis. The green (510–530 nm, FL-1) and red (650 nm, FL-3) fluorescence of AO, following blue (488 nm) excitation, was determined for 10,000 events and measured on a FACScan cytofluorimeter using Cell Quest software (Becton Dickinson).

### 4.11. Immunohistochemical Staining

For immunohistochemical assessments, 4-μm-thick paraffin-embedded OS sections were mounted on coated glass slides to detect the proteins under investigation. Following antigen retrieval and blocking of endogenous peroxidases and nonspecific protein binding, slide sections were incubated first with primary antibodies, followed by appropriate horseradish peroxidase-conjugated secondary antibodies. Primary antibodies against p-Akt, Ki67, LC3, Beclin1, and p-mTOR (diluted 1:200) were purchased from Cell Signaling Technology. All slides were developed with 3,3′ diaminobenzidine, followed by hematoxylin counterstaining.

### 4.12. Terminal Deoxynucleotidyl Transferase-Mediated dUTP Nick-End Labeling (TUNEL) Assays

Mice from the control and ZOL-treated groups were used for the in vivo apoptosis study. Tumors were collected and fixed with 10% neutral-buffered formalin. Deparaffinized sections were incubated with 20 μg/mL protease K for 15 min at room temperature, washed with PBS, and incubated with the TUNEL reaction mixture (Millipore, Burlington, MA, USA) for 1 h at 37 °C in a humidified chamber. Tissue sections were then incubated with an anti-digoxigenin–peroxidase mixture and subsequently with the peroxidase substrate. Images were acquired using a Nikon Eclipse Ts2R-FL microscope.

### 4.13. Elisa Assay

4EBP1 and p70S6K activities were measured using Elisa assay kits (Cell signaling technology) according to the manufacturer’s recommendations. Data were collected using a Multiskan EX at 450 nm [49].

### 4.14. miRNA and Transient Transfection

The control mimics, miR-212-3p mimics, miR-212-3p inhibitors, and control inhibitors were all purchased from Bioneer (Daejeon, Korea). Cells were transiently transfected with 60 nM control or miR-212-3p mimics using G-fectin miRNA Transfection Reagent [50].

### 4.15. Quantitative Reverse-Transcriptase Polymerase Chain Reaction (qRT-PCR) Experiments

Total RNA was obtained manually using TRIzol (Invitrogen, Carlsbad, CA, USA). All qRT-PCR were performed using the KAPA SYBR FAST qPCR Kit from KAPA Biosystems (Wilmington, MA, USA), according to the manufacturer’s instructions. Reactions were carried out in a Rotor Gene Q instrument (Qiagen, Seoul, Korea), and results were expressed as fold-changes in expression, calculated using the ΔΔCt method relative to the control sample. β-actin was used as an internal normalization control. The following primer pairs were employed for amplifying Beclin1: 5′-AATTGTGAGGACACCCAAGC-3′ (sense) and 5′-AGGTTGAGAAAGGCGAGACA-3′ (antisense).

### 4.16. In Vivo Tumor Model and Administration of ZOL and miR-212-3p Mimic

Therapy began when the tumor volume reached ~150 mm^3^. The tumor-bearing mice were randomly assigned into five groups (*n* = 4): (1) Control mice receiving intraperitoneal (i.p.) injection of 100 μL PBS and intratumoral injection of 10 μg miRNA negative mimic; (2) mice receiving i.p. injection of ZOL (0.1 mg/kg) in 100 μL PBS; (3) mice receiving intratumoral injection of 10 μg has-miR-212-3p mimic; (4) mice receiving i.p. injection of ZOL (0.1 mg/kg) in 100 μL PBS and intratumoral injection of 10 μg has-miR-212-3p mimic; (5) mice receiving i.p. injection of ZOL (0.1 mg/kg) and intratumoral injection of 10 μg has-miR-212-3p inhibitor. For in vivo administration of miRNA mimic, the negative control, miR-212-3p mimic, or has-miR-212-3p inhibitor was complexed with in vivo-JetPEI at an N/P ratio of 6 in 5% glucose solution (a total of 100 μL) for intratumoral injection. The mice were treated twice a week.

### 4.17. Positron Emission Tomography (PET)/Computed Tomography (CT) Acquisition

PET/CT acquisitions were performed with a NanoScan^®^ PET/CT (Mediso Medical Imaging Systems, Budapest, Hungary). Mice were fasted for 12 h before image acquisition. The mice were anesthetized with a mixture of 2% isoflurane and oxygen. The mice were injected via the tail vein with 200 µCi of 18F-fluoro-2-deoxy-d-glucose ([^18^F]-FDG) in 200 μL of saline. Before PET imaging, CT images were acquired for 2.5 min for a structural reference and attenuation correction using 50 kVp of X-ray voltage and 0.16 mAs of anode current. At 60 min after [^18^F]-FDG injection, static PET images were acquired for 20 min with the 400–600 keV energy window. All obtained PET images were reconstructed with 0.86 × 0.86 × 0.80 mm^3^ of voxel dimensions using four iterations of the 3-dimensional ordered subset expectation maximization (3D-OSEM) algorithm with six subsets.

To quantify [^18^F]-FDG uptake, PMOD software (version 3.8, PMOD Group, Graubünden, Switzerland) was used to calculate the standardized uptake value (SUV). The SUV of each voxel was defined as follows: Radioactivity of each voxel was multiplied by the mouse weight and divided by the injected dose. The volumes of interest (VOI) were manually defined for the tumor, and then SUV_max_ and SUV_mean_ were calculated in the VOI. SUV_max_ is the maximum concentration of tracer within the VOI, and SUV_mean_ is the mean concentration of tracer within the VOI.

### 4.18. Statistical Analysis

Statistical significance was determined using Student’s *t*-test. Differences were considered significant at *p* ≤ 0.05 or 0.001.

### 4.19. Data Availability

The data supporting this study are available from the corresponding authors on reasonable request.

## 5. Conclusions

In conclusion, drug-induced, miRNA-regulated autophagy is increasingly becoming a therapeutic strategy for managing treatment of cancer patients. Our findings provide insights into the anticancer efficacy of ZOL and miR-212-3p, supporting a preclinical rationale for conducting a large-scale clinical study on treating OS.

## Figures and Tables

**Figure 1 cancers-11-01812-f001:**
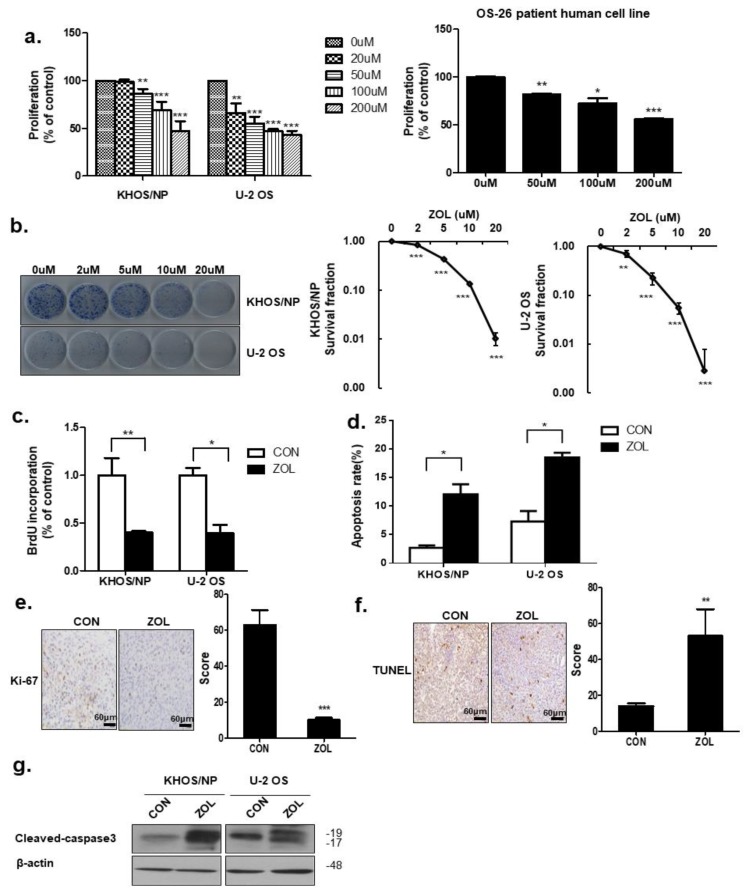
Zoledronic acid (ZOL) inhibited osteosarcoma (OS) cell proliferation. (**a**) Cell viability was evaluated by MTT assay in KHOS/NP cells, U2OS cells of OS cell lines and cells from a patient with OS after 48 h treatment with the indicated concentration of ZOL; * *p* < 0.05, ** *p* < 0.01, *** *p* < 0.001. (**b**) Colony-formation assays were performed using KHOS/NP and U2OS cells treated with the indicated concentration of ZOL for seven days; ** *p* < 0.01, *** *p* < 0.001. (**c**) Cells were treated with ZOL (40 μM) for 72 h, and the proliferation rate was detected by 5-bromo-2′-deoxyuridine (BrdU) labeling; * *p* < 0.05, ** *p* < 0.01. (**d**) The apoptosis rate was assessed by fluorescence-activated cell sorting (FACS) analysis for 72 h treatment; * *p* < 0.05. (**e**) Ki67 expression in the orthotopic model was examined by immunohistochemistry; *** *p* < 0.001. (**f**) Two weeks after tumor cell inoculation, mice were randomly assigned into four groups of three animals each: Control group (untreated), ZOL alone group. ZOL was administered intraperitoneally twice weekly at a dose of 0.1 mg/kg in 100 μL PBS two weeks after inoculation. TUNEL assays were performed using orthotopic cells [23]; ** *p* < 0.01. (**g**) Immunoblotted cell lysates (30 μg) are shown with the cleaved caspase3 and β-actin antibodies for 48 h treatment.

**Figure 2 cancers-11-01812-f002:**
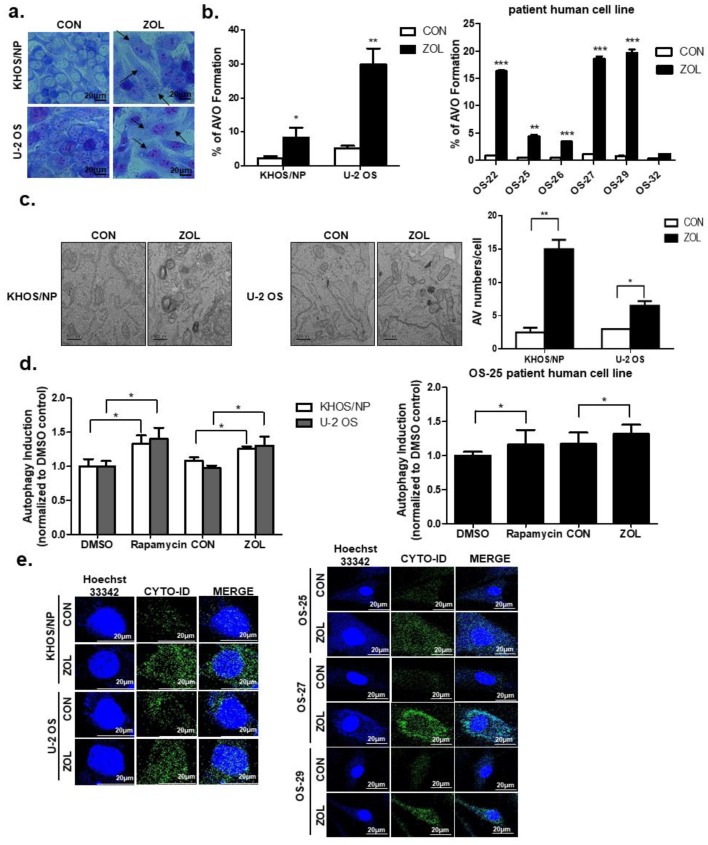
Zoledronic acid (ZOL) induced accumulation of acidic vacuoles (AVOs). (**a**) Cells were stained with Giemsa (10% in PBS), washed, and imaged under a Nikon Eclipse Ts2R-FL microscope (magnification, 40×). Black arrows point to vacuoles. A representative image from two independent experiments is shown. (**b**) Cells were treated with ZOL for 48 h and then stained with acridine orange. Green and red fluorescence in acridine orange (AO)-stained cells was detected by flow cytometry; * *p* < 0.05, ** *p* < 0.01, *** *p* < 0.001. (**c**) Autophagy measured by TEM in ZOL-treated OS cells (left). The quantification was added in (**b**) (right); * *p* < 0.05, ** *p* < 0.01. (**d**,**e**) Cells were treated with rapamycin (4 μM) for 18 h and ZOL (80 μM) for 48 h to detect the CYTO-ID^®^ dye signal; * *p* < 0.05.

**Figure 3 cancers-11-01812-f003:**
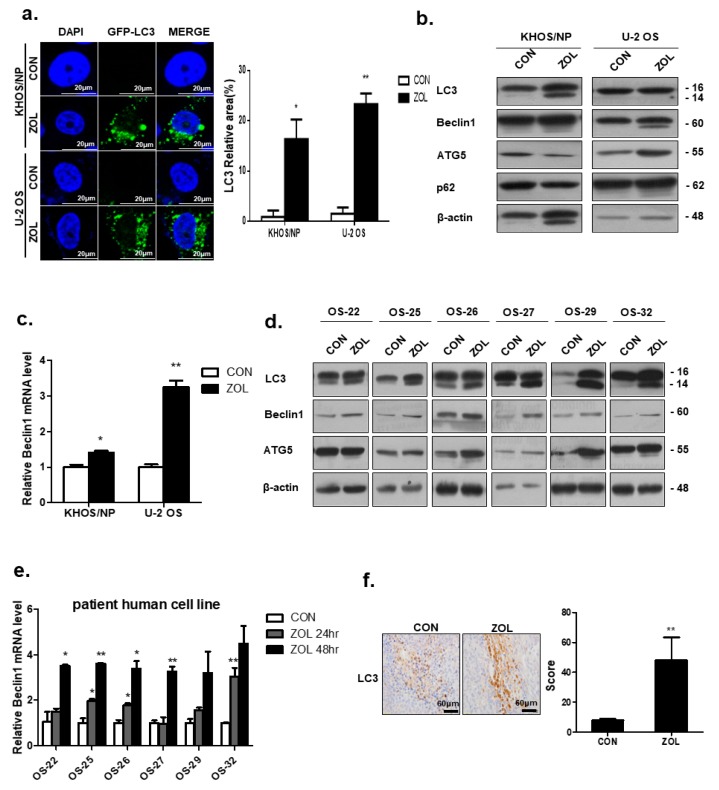
Zoledronic acid (ZOL) induced autophagy in osteosarcoma (OS) cells and patient-derived OS cells. (**a**) Induction of autophagy in ZOL-treated KHOS/NP and U2OS cells with stable expression of Green Fluorescent Protein (GFP)-tagged LC3 (left). The quantification was added in (**a**) (right); **p* < 0.05, ***p* < 0.01. (**b**,**c**) Immunoblotting of LC3, Beclin-1, ATG5, and p62 and qRT-PCR analysis of Beclin1 mRNA level in KHOS/NP and U2OS cells treated with ZOL for 48 h; * *p* < 0.05, ** *p* < 0.01. (**d**,**e**) Immunoblotting of LC3, Atg5, and Beclin-1 and qRT-PCR analysis of Beclin1 mRNA level in patient-derived OS cells that were treated with ZOL.; * *p* < 0.05, ** *p* < 0.01. (**f**) LC3 expression in an orthotopic model was examined by immunohistochemistry. Representative images are provided, as indicated; ** *p* < 0.01.

**Figure 4 cancers-11-01812-f004:**
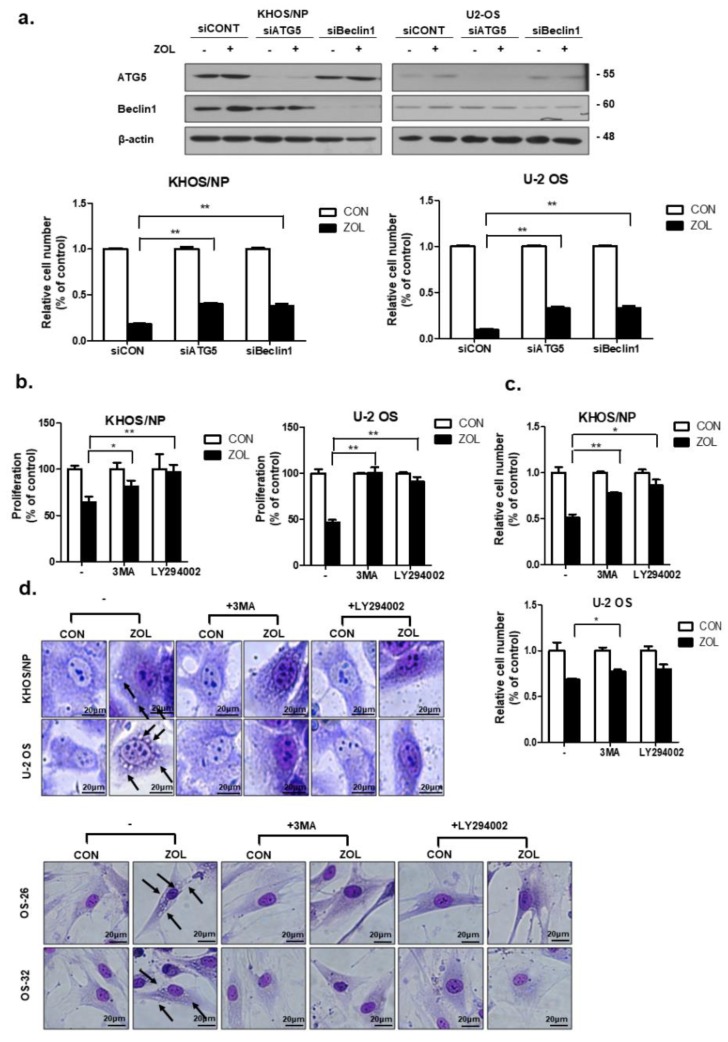
Inhibition of autophagy repressed the anti-proliferative effect of zoledronic acid (ZOL) in osteosarcoma (OS) cells. (**a**) Cells were transfected with si-ATG5 or Beclin-1 or a control siRNA (40 nM) for 24 h, after which the cells were treated with ZOL for another 48 h. The proliferation rate was detected using trypan blue cell-counting assays and immunoblotting was conducted to check the efficiency of transfection; ***p* < 0.01. (**b**,**c**) Cells were treated with 3MA and LY294002 in the presence or absence of ZOL for 48 h, and the proliferation rate was measured by MTT and trypan blue cell-counting assays; * *p* < 0.05, ** *p* < 0.01. (**d**) Cells were grown in six-well tissue culture plates under standard cell culture conditions, pre-treated (24 h) with 3-MA (2 mM) and LY294002 (20 μM), and then treated with ZOL for 48 h. Cells were stained with Giemsa (10% in PBS), washed, and photographed under a Nikon Eclipse Ts2R-FL microscope (magnification 40×). Black arrows show vacuoles. A representative image from two independent experiments is shown.

**Figure 5 cancers-11-01812-f005:**
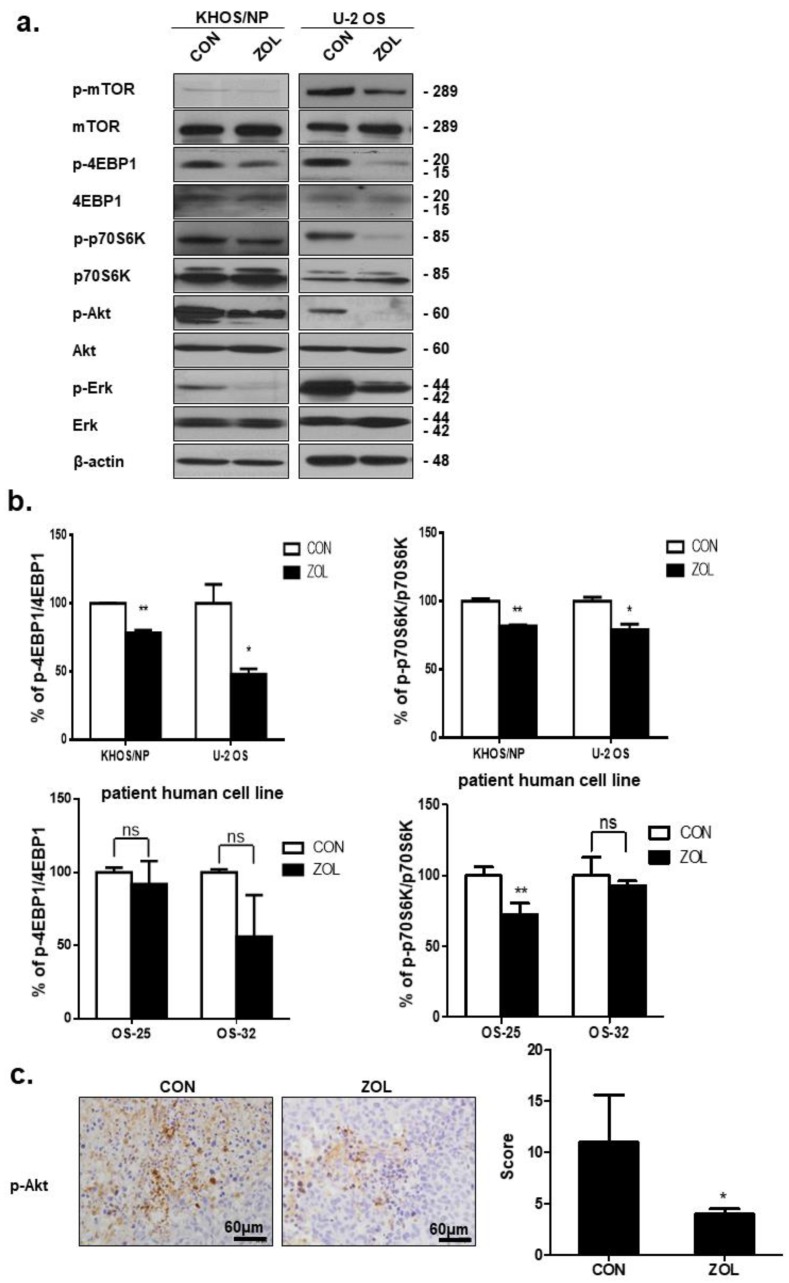
Zoledronic acid (ZOL) induced autophagy by repressing the Akt/mTOR pathway in osteosarcoma (OS) cells and in an orthotopic in vivo model. (**a**) Immunoblotted cell lysates (30 μg) are shown with the corresponding antibodies. (**b**) ELISA was performed to quantify the level of phosphor-p70S6K at Thr389, phosphor-4EBP1 at Thr37/46, and p70S6K-4EBP1 in KHOS/NP, U2OS, and cells from a patient with OS after ZOL treatment. Each concentration was tested in quadruplicate, and each experiment was repeated two times. The data shown represent the combined mean ± SD from two independent experiments; * *p* < 0.05, ** *p* < 0.01, ns > 0.05. (**c**) *p*-Akt expression in the orthotopic model was examined by immunohistochemistry. Representative images are provided as indicated; * *p* < 0.05.

**Figure 6 cancers-11-01812-f006:**
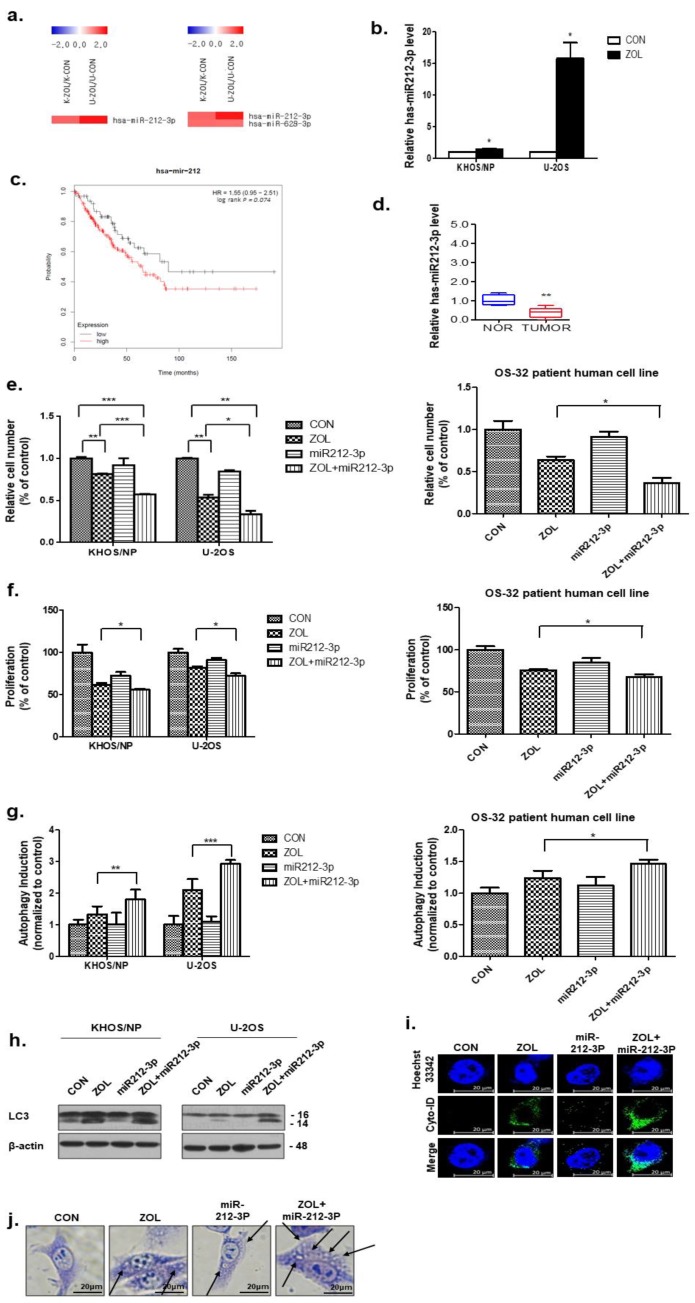
miR-212-3p directly targets autophagy in zoledronic acid (ZOL)-treated osteosarcoma (OS) cells. (**a**) Expression analysis of miRNAs upregulated after ZOL treatment. (**b**) miR-212-3p levels were analyzed by qRT-PCR in OS cells treated with ZOL. * *p* < 0.05 (**c**) Kaplan–Meier survival curves for sarcoma patients based on miR-212-3p expression. The survival rate is shown; *p* < 0.05. (**d**) The relative expression of miR-212-3p in matched primary OS tissues and non-tumor tissues. (**e**) Cells were treated with ZOL or miR-212-3p for 48 h, and the proliferation rate was measured by cell counting; * *p* < 0.05, ** *p* < 0.01, *** *p* < 0.001. (**f**) The proliferation rate was detected by MTT assay at the same time point. * *p* < 0.05. (**g**) Two OS cell lines and patient-derived cells were treated with ZOL (80 μM) and miR-212-3p or a combination for 48 h. ZOL + miR-212-3p treatment resulted in an increase in the CYTO-ID^®^ dye signal compared to that following either single treatment; * *p* < 0.05, ** *p* < 0.01, *** *p* < 0.001. (**h**) Immunoblotting of LC3 in lysates from KHOS/NP and U2OS cells. (**i**) ZOL + miR-212-3p treatment resulted in an increase in the CYTO-ID^®^ dye signal in KHOS/NP cells after 48 h. (**j**) Cells were treated with ZOL or miR-212-3p for 48 h, stained with Giemsa (10% in PBS), washed, and photographed under an Eclipse Ts2R-FL microscope (magnification 40×). The black arrows point to vacuoles. A representative image from two independent experiments is shown.

**Figure 7 cancers-11-01812-f007:**
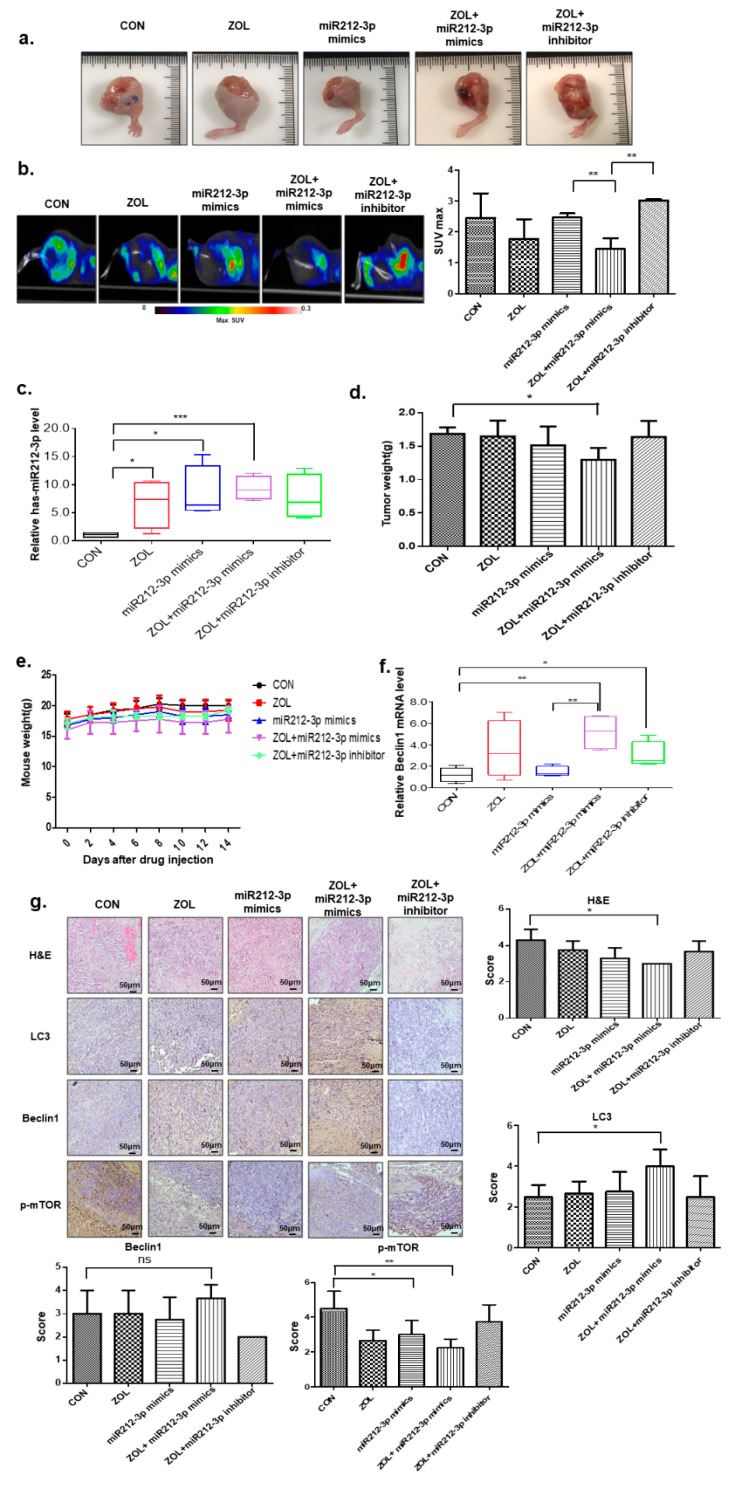
The autophagic target relationship between miR-212-3p and zoledronic acid (ZOL) in an in vivo model. (**a**) Image of isolated tumors derived from osteosarcoma (OS) xenografts intratumorally treated with ZOL or miR- 212-3p mimics or inhibitor. (**b**) Representative PET/CT images of KHOS tumor-bearing mice after injection of [^18^F]-[Fluorine-18(18F)]-fluorodeoxyglucose (FDG). The radioactivity of [^18^F]-FDG in tumors is presented as the maximal value of SUV (mean ± S.D); ** *p* < 0.01. (**c**) miR-212-3p levels were analyzed by qRT-PCR in in vivo tissues treated with ZOL only; miR212-3p mimics only; ZOL + miR212-3p mimics; or ZOL + miR212-3p inhibitor. Values represent the means of three experiments ± SD; * *p* < 0.05, *** *p* < 0.001. (**d**) Tumors were excised and weighed at the end of the experiment (six weeks after tumor cell inoculation); * *p* < 0.05. (**e**) Mouse body weights were assessed at 14 days. (**f**) Beclin1 mRNA expression levels in mice receiving each treatment; * *p* < 0.05, ** *p* < 0.01, *** *p* < 0.001. (**g**) Hematoxylin and eosin (H&E) staining and LC3, Beclin1, and *p*-mTOR expression in tumor xenografts were examined by immunohistochemistry. * *p* < 0.05, ** *p* < 0.01, ns > 0.05.

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
