# Peer review of "Synergistic Autophagy Effect of miR-212-3p in Zoledronic Acid-Treated In Vitro and Orthotopic In Vivo Models and in Patient-Derived Osteosarcoma Cells"

_cancers, 2019, doi:10.3390/cancers11111812_

Round 1

Reviewer 1 Report

Ho Kim et al., investigate the effect of ZOL treatment in OS cell proliferation showing that autophagy play a pivotal role. The paper is well written and several experiments have been performed, however major points should be adressed by the authors prior publication:

Fig1a: Please indicate the duration of ZOL treatment.

Fig1b: Please explain why different concentrations of ZOL where used compared to Fig1a.

Fig1a and Fig1b: Statistical analysis is missing. Please add. Moreover, I would like to suggest to check all the figure and when is not significant different please add ns.

Fig1f: Please indicate the concentration and duration of ZOL treatment.

Fig1e and Fig1f: Which type of cell death pathway is activated by ZOL? Could you please use more specific markers such as Caspase-3?

Fig. 2c: Quantification of autophagosome formation by TEM is missing. Try to be more precise and explain which was the treatment used for this experiment.

Fig3a and 3b: and 3d Please provide quantification for GFP-LC3 puncta and WB in 3b and 3d, including LC3 for all cell lines tested (not only beclin1).

Fig4a. Please provide WB showing the efficiency of ATG5 and Beclin1 KD. The restoration of cell number is quite small, can be due to a limited KD efficiency? Please discuss and I would like to suggest to try both siRNAs together.

Fig.4: Data showing that inhibition of autophagy repressed ZOL effect are not very convincing due to a very small effect and the variability of different cell lines. Please add in the text a sentence explaining how other pathways can be involved.

Fig5: Please indicate significant in all graphs. If not significant add: ns.

Fig. 7: Tumor weight seems to reduce at all!!! Please measure significant.

Author Response

Fig1a: Please indicate the duration of ZOL treatment.

[Answer] As the reviewer recommended, we have added missing information.

Fig1b: Please explain why different concentrations of ZOL where used compared to Fig1a.

[Answer] In order to colonize the cells, the drug was incubated for about 7 days or more, so the high dose of the figure 1a was not used, so CFA was performed under this condition.

Fig1a and Fig1b: Statistical analysis is missing. Please add. Moreover, I would like to suggest to check all the figure and when is not significant different please add ns.

[Answer] As the reviewer recommended, we have added missing significance.

Fig1f: Please indicate the concentration and duration of ZOL treatment.

[Answer] As the reviewer recommended, we have added missing information.

Fig1e and Fig1f: Which type of cell death pathway is activated by ZOL? Could you please use more specific markers such as Caspase-3?

[Answer] As the reviewer recommended, requested experiments were added as western blotting in the revised manuscript.

Fig. 2c: Quantification of autophagosome formation by TEM is missing. Try to be more precise and explain which was the treatment used for this experiment.

[Answer] As the reviewer recommended, requested quantification were added in the revised figure.

Fig3a and 3b: and 3d Please provide quantification for GFP-LC3 puncta and WB in 3b and 3d, including LC3 for all cell lines tested (not only beclin1).

[Answer] As the reviewer recommended, requested quantification and indicated WB blots were added in the revised manuscript.

Fig4a. Please provide WB showing the efficiency of ATG5 and Beclin1 KD. The restoration of cell number is quite small, can be due to a limited KD efficiency? Please discuss and I would like to suggest to try both siRNAs together.

[Answer] As the reviewer recommended, requested experiments were did as following and described this part in the result and figure 4a sections.

Fig.4: Data showing that inhibition of autophagy repressed ZOL effect are not very convincing due to a very small effect and the variability of different cell lines. Please add in the text a sentence explaining how other pathways can be involved.

[Answer] As the reviewer recommended, we have explained the following contents in discussion section.

Fig5: Please indicate significant in all graphs. If not significant add: ns.

[Answer] As the reviewer recommended, we have added missing significance.

Fig. 7: Tumor weight seems to reduce at all!!! Please measure significant.

[Answer] As the reviewer recommended, we have added missing information.

Reviewer 2 Report

In this manuscript, the authors aimed to clarify the mechanism through which Zoledronic acid inhibits OS cell proliferation and the role exerted by autophagic process that, as it is known, can play opposite roles in cancer cells. The paper is the natural continuation of a previous study by the same authors on ZOL and IR. An interesting aspect of the manuscript is represented by the synergistic effect exerted by the combination between ZOL and miR-212-3p even if, in my opinion, the field of miRNAs is really delicate, since the results can create false positive/negative data.

The paper presents solid results, the experimental procedures are reasonable and the conclusions coherent with the obtained results. Some results need to be presented in a clearer way.

Detailed comments and suggestions:

The Authors should better comment on the results obtained. For example, to discuss the different sensitivity of cells to the compound;

They state that ZOL induces autophagy in both cell lines and primary OS cells, but by a careful analysis of the results of western blotting, not all proteins seem to vary after treatment (ATG5, LC3 in OS-29 or OS-32). In this regard it is necessary to report the densitometric analysis of the western blotting bands and to evaluate their statistical significance.

Fig. 1a, d, e: the legend does not report the time of treatment

Fig 2b: it is not clear the y axis (% of cell acridine orange), please specify;

Fig. 2d: it is reported the effect of rapamycin but the result is not discussed in the text.

Fig. 2 legend (lane 136-137): it is reported a conclusion that it is not necessary in a legend.

Fig. 3, legend (b, c): it is reported qRT-PCR analysis of all autophagic markers but the Figure 3c shows only Beclin mRNA.

Fig. 5b: It is not clear the type of ELISA procedure

Fig. 6f: the authors employed different doses of ZOL in the experiments. It is opportune to specify the choice, for example, of 80 mM ZOL when employed in combination with miRna. Generally, in combination lower doses of the single compounds are employed, just to demonstrate synergistic effects. Please, specify

Fig. 6h: the figure does not show cell lysates with the corresponding antibodies, but the levels of the autophagic markers in cell lysates

Paragraph 2.7, lane 240: to explore this phenotype, please specify

Paragraph 4.13: The transfection procedure is not reported. Please to specify or report a Reference

 Minor comments:

1: to adjust micromolar μM Lane 112: the sentence is not clear, please modify 4a: to correct siATG5 Paragraph 4.10, lane 420: cells were treated with various concentrations of ZOL (40 μM), please modify because it is a single dose.

Author Response

Fig. 1a, d, e: the legend does not report the time of treatment

[Answer] As the reviewer recommended, we have added missing information.

Fig 2b: it is not clear the y axis (% of cell acridine orange), please specify;

[Answer] As the reviewer recommended, we have corrected as % AVO formation as FACS analysis.

Fig. 2d: it is reported the effect of rapamycin but the result is not discussed in the text.

[Answer] As the reviewer recommended, we have added missing description.

Fig. 2 legend (lane 136-137): it is reported a conclusion that it is not necessary in a legend.

[Answer] As the reviewer recommended, we have removed following description in the Fif2 legend.

Fig2 legend: ZOL treatment resulted in an increase in the CYTO-ID® dye signal in a sample from a patient with OS.

Fig. 3, legend (b, c): it is reported qRT-PCR analysis of all autophagic markers but the Figure 3c shows only Beclin mRNA.

[Answer] We confirmed only Beclin mRNA. It is known that LC3 mRNA is poorly used as an autophagic markers and LC3 is also detected on secretory lysosomes in osteoblast-like cells. Thus, we removed the data to avoid these discrepancies.

Fig. 5b: It is not clear the type of ELISA procedure

[Answer] As the reviewer recommended, we have added missing reference and protocol.

Fig. 6f: the authors employed different doses of ZOL in the experiments. It is opportune to specify the choice, for example, of 80 mM ZOL when employed in combination with miRna. Generally, in combination lower doses of the single compounds are employed, just to demonstrate synergistic effects. Please, specify

[Answer] In our previous publication, the merged condition with mirna is synonymous with the single processing condition. Therefore, in this paper, the same concentration was used as the single condition when combining. We would very appreciate it if you understand this.

Ref) Eun Ho Kim et al. Tumor treating fields induce autophagy by blocking the Akt2/miR29b axis in glioblastoma cells. Oncogene volume 38, pages6630–6646(2019)

Fig. 6h: the figure does not show cell lysates with the corresponding antibodies, but the levels of the autophagic markers in cell lysates

[Answer] We apologize for not understanding your meaning in detail. As our result, ZOL plus miR-212-3p treatment resulted in marked autophagy induction, as evidenced by increased LC3-II conversion and LC3 puncta. If you describe in detail, we will revise as your comment.

Paragraph 2.7, lane 240: to explore this phenotype, please specify.

[Answer] As the reviewer recommended, we have added the spectrum in the figure to show the phenotype.

Paragraph 4.13: The transfection procedure is not reported. Please to specify or report a Reference

[Answer] As the reviewer recommended, we have added missing reference and protocol.

 Minor comments:

1: to adjust micromolar μM Lane 112: the sentence is not clear, please modify 4a: to correct siATG5 Paragraph 4.10, lane 420: cells were treated with various concentrations of ZOL (40 μM), please modify because it is a single dose.

[Answer] As the reviewer recommended, we have revised the description.

Reviewer 3 Report

The authors investigated the role of autophagy induction in the zoledronic acid (ZOL)-mediated suppression of human osteosarcoma (OS) cell proliferation. They found that ZOL induces autophagy in association with cell death rather than cell survival in OS cells. Moreover, miR-212-3p activation enhanced the ZOL-induced autophagy and antitumor effect in OS cells. Although these findings are potentially valuable, these results are not conclusive. There are some points that require clarification before the manuscript can be considered suitable for publication.

Major issues:

Figures 2 and 3: The authors demonstrated that ZOL induces autophagy, which is confirmed by accumulation of acidic vacuoles, LC3-II, and Beclin1, in OS cells. However, Atg5 expression was not increased in the ZOL-treated OS cells. To further confirm the autophagy induction by ZOL treatment, the authors should show p62 downregulation, which is also autophagy-related marker.   Figure 4: The authors demonstrated that both siATG5 and siBeclin1 attenuated the ZOL-mediated suppression of OS cell viability compared to siControl. The authors should show the WB data for siATG5 and siBeclin1-mediated downregulation of ATG5 and Beclin1 protein compared to siControl.   Figure 5: The authors demonstrated that ZOL suppresses the mTOR/p70S6K signaling pathway. The authors should show the WB data for non-phosphorylated mTOR, 4EBP1, p70S6K, Akt, and Erk proteins.   Figure 6: The authors demonstrated that miR-212-3p mimic enhances the ZOL-mediated autophagy induction in OS cells. However, miR-212-3p mimic alone did not induce autophagy in OS cells, suggesting that miR-212-3p activation is not involved in the ZOL-mediated autophagy induction. The authors should confirm the molecular mechanism in the miR-212-3p-mediated enhancement of ZOL-induced autophagy, such as the possible target genes for miR-212-3p.   Figure 7d: The authors demonstrated that tumor weight was reduced in combination-treated mice compared to single treated group (Page 14, line 249). However, there was no significant difference in the graph. The authors should show the significant difference by multi-group comparison.  

Minor issues:

Figure 2a: two photographs for control KHOS/NP and U-2OS were similar. The authors should correct them. Figure 5c: The number of pAkt-positive cells seems to be higher in ZOL-treated tumors than control tumors. However, the p-Akt score was higher in control tumors than ZOL-treated tumors. The authors should show the representative photographs.

Author Response

Major issues:

Figures 2 and 3: The authors demonstrated that ZOL induces autophagy, which is confirmed by accumulation of acidic vacuoles, LC3-II, and Beclin1, in OS cells. However, Atg5 expression was not increased in the ZOL-treated OS cells. To further confirm the autophagy induction by ZOL treatment, the authors should show p62 downregulation, which is also autophagy-related marker.  

[Answer] As the reviewer recommended, requested experiments were added in the revised figure 3b and manuscript.

Figure 4: The authors demonstrated that both siATG5 and siBeclin1 attenuated the ZOL-mediated suppression of OS cell viability compared to siControl. The authors should show the WB data for siATG5 and siBeclin1-mediated downregulation of ATG5 and Beclin1 protein compared to siControl.  

[Answer] As the reviewer recommended, requested experiments were added in the revised figure 4a and manuscript.

Figure 5: The authors demonstrated that ZOL suppresses the mTOR/p70S6K signaling pathway. The authors should show the WB data for non-phosphorylated mTOR, 4EBP1, p70S6K, Akt, and Erk proteins.

[Answer] As the reviewer recommended, requested experiments were added in the revised figure and manuscript.

 Figure 6: The authors demonstrated that miR-212-3p mimic enhances the ZOL-mediated autophagy induction in OS cells. However, miR-212-3p mimic alone did not induce autophagy in OS cells, suggesting that miR-212-3p activation is not involved in the ZOL-mediated autophagy induction. The authors should confirm the molecular mechanism in the miR-212-3p-mediated enhancement of ZOL-induced autophagy, such as the possible target genes for miR-212-3p.  

[Answer] As the reviewer recommended, we have explained the following contents in the discussion section.

MiR-212 is a novel cancer-related miRNA and is known to be involved in the development of various types of human cancers and has been shown to reduce miR-212 expression in renal cell carcinoma and ovarian cancer. In general, miRNA is known to inhibit translation by targeting genes. Pengbo Jia et al, found that miR212 targets FOXM1 and inhibits its expression by inhibiting the Wnt / b-catenin signaling pathway, thereby inhibiting migration and tumorigenicity in hepatocellular carcinoma.
In addition, C.ZHOU et al. proved that miR212 inhibited the expression of TCF7L2, thereby inhibit proliferation and metastasis in cervical cancer. As following reference, we will uncover the targets of miR-212-3p to reveal multiple cellular functions related to ZOL treatment.

Ref) Upregulation of MiR-212 Inhibits Migration and Tumorigenicity and Inactivates

Wnt/b-Catenin Signaling in Human Hepatocellular Carcinoma Technol Cancer Res Treat. 2018 Jan 1;17:1533034618765221.

Ref) Effect of miR-212 targeting TCF7L2 on the proliferation and metastasis of cervical cancer Eur Rev Med Pharmacol Sci. 2017 Jan;21(2):219-226.

Figure 7d: The authors demonstrated that tumor weight was reduced in combination-treated mice compared to single treated group (Page 14, line 249). However, there was no significant difference in the graph. The authors should show the significant difference by multi-group comparison.  

[Answer] As the reviewer recommended, mice with large deviations in tumor volume were excluded from the calculations to regain significance.

Minor issues:

Figure 2a: two photographs for control KHOS/NP and U-2OS were similar. The authors should correct them.

[Answer] We apologize for the error. We have changed to new data in Figure.

Figure 5c: The number of pAkt-positive cells seems to be higher in ZOL-treated tumors than control tumors. However, the p-Akt score was higher in control tumors than ZOL-treated tumors. The authors should show the representative photographs.

[Answer] We have replaced the previous data with clearer data. And we also have included a corresponding detailed description in the Results section.

Reviewer 4 Report

This research paper, Synergistic autophagy effect of miR 212-3p in zoledronic acid-treated in vitro and orthotopic in vivo models and in patient-derived osteosarcoma cells, details the results section. This study is also with significance of content.

Author Response

Reviewer4) This research paper, Synergistic autophagy effect of miR 212-3p in zoledronic acid-treated in vitro and orthotopic in vivo models and in patient-derived osteosarcoma cells, details the results section. This study is also with significance of content.

[Answer] We are sincerely appreciative of the acceptance of our manuscript for publication in your journal.

Round 2

Reviewer 1 Report

The authors adressed all my concerns.

Author Response

Reviewer1) The authors adressed all my concerns.

[Answer] We are sincerely appreciative of the acceptance of our manuscript for publication in your journal.

Reviewer 2 Report

The Authors responded to the comments. The manuscript can be accepted after two small corrections here reported.

Comment:  Fig. 3, legend (b, c): it is reported qRT-PCR analysis of all autophagic markers but the Figure 3c shows only Beclin mRNA.

[Answer] We confirmed only Beclin mRNA. It is known that LC3 mRNA is poorly used as an autophagic markers and LC3 is also detected on secretory lysosomes in osteoblast-like cells. Thus, we removed the data to avoid these discrepancies.

About this comment,  we recommend replacing the sentence in the legend of Fig. 3b and c with

Immunoblotting of LC3, Atg5, and Beclin-1 164 and qRT-PCR analysis of Beclin1 mRNA level in patient-derived OS cells that were treated with ZOL.

Comment: Fig. 6h: the figure does not show cell lysates with the corresponding antibodies, but the levels of the autophagic markers in cell lysates

[Answer] We apologize for not understanding your meaning in detail. As our result, ZOL plus miR-212-3p treatment resulted in marked autophagy induction, as evidenced by increased LC3-II conversion and LC3 puncta. If you describe in detail, we will revise as your comment.

About this comment, we recommend replacing the sentence in the legend of Fig. 6h with

Immunoblotting of LC3 in lysates from KHOS/NP and U2OS cells.

Author Response

Reviewer2)

The Authors responded to the comments. The manuscript can be accepted after two small corrections here reported.

Comment:  Fig. 3, legend (b, c): it is reported qRT-PCR analysis of all autophagic markers but the Figure 3c shows only Beclin mRNA.

[Answer] We confirmed only Beclin mRNA. It is known that LC3 mRNA is poorly used as an autophagic markers and LC3 is also detected on secretory lysosomes in osteoblast-like cells. Thus, we removed the data to avoid these discrepancies.

About this comment,  we recommend replacing the sentence in the legend of Fig. 3b and c with

Immunoblotting of LC3, Atg5, and Beclin-1 164 and qRT-PCR analysis of Beclin1 mRNA level in patient-derived OS cells that were treated with ZOL.

[Answer] As the reviewer recommended, we have changed description.

Comment: Fig. 6h: the figure does not show cell lysates with the corresponding antibodies, but the levels of the autophagic markers in cell lysates

[Answer] We apologize for not understanding your meaning in detail. As our result, ZOL plus miR-212-3p treatment resulted in marked autophagy induction, as evidenced by increased LC3-II conversion and LC3 puncta. If you describe in detail, we will revise as your comment.

About this comment, we recommend replacing the sentence in the legend of Fig. 6h with

Immunoblotting of LC3 in lysates from KHOS/NP and U2OS cells.

[Answer] As the reviewer recommended, we have changed description.

Reviewer 3 Report

The authors addressed the appropriate comments by showing the additional data in the revised Figures.

Author Response

Reviewer3) The authors addressed the appropriate comments by showing the additional data in the revised Figures.

[Answer] We are sincerely appreciative of the acceptance of our manuscript for publication in your journal.

Round 3

Reviewer 2 Report

The manuscript is accept in this form